# WARM STARTS ACCELERATE CONDITIONAL DIFFUSION

## ABSTRACT

Generative models like diffusion and flow-matching create high-fidelity samples by progressively refining noise. The refinement process is notoriously slow, often requiring hundreds of function evaluations. We introduce *Warm-Start Diffusion* (WSD), a method that uses a simple, deterministic model to dramatically accelerate *conditional* generation by providing a better starting point. Instead of starting generation from an uninformed $\mathcal{N}(\mathbf{0}, I)$ prior, our deterministic warm-start model predicts an informed prior $\mathcal{N}(\hat{\boldsymbol{\mu}}_C, \mathrm{diag}(\hat{\boldsymbol{\sigma}}_C^2))$, whose moments are conditioned on the input context $C$. This *warm start* substantially reduces the distance the generative process must traverse, and therefore the number of diffusion steps required when the context $C$ is strongly informative. WSD is applicable to any standard diffusion or flow-matching algorithm, is orthogonal to and synergistic with other fast sampling techniques like efficient solvers, and is simple to implement. We test WSD in a variety of settings, and find that it substantially outperforms standard diffusion in the efficient sampling regime, generating realistic samples using only 4-6 function evaluations, and saturating performance with 10-12.

## 1 INTRODUCTION

Generative models based on stochastic processes, like diffusion and flow-matching, have become the state of the art for high-fidelity data synthesis (Ho et al., 2020; Song et al., 2020; Karras et al., 2022).

Despite the success of diffusion, its practical application is often limited by a significant bottleneck: slow, iterative sampling that can require a Number of Function Evaluations (NFE) in the hundreds to generate a single sample. This cost becomes particularly problematic in domains where each sample is itself only part of an autoregressive rollout that can contain hundreds or thousands of samples, highlighting the importance of computationally efficient methods for conditional diffusion. Our work focuses on accelerating sampling for this class of problems.

Significant progress has been made from the inefficient foundational DDPM method (Ho et al., 2020) that required $\sim 1000$ steps per sample: Re-framing the diffusion process in a continuous-time setting opened the door for much faster sampling (Song et al., 2020). Subsequent methods have further reduced the step count by developing more efficient ways to solve the underlying ordinary differential equation (ODE). These advancements include deterministic samplers like DDIM (Song et al., 2022), which enabled larger step sizes; higher-order numerical solvers like DPM-Solver(++) (Lu et al., 2022; 2025), which approximate the ODE solution more accurately per step; and novel training paradigms like flow matching (Lipman et al., 2022), which aim to learn simpler, straighter generative paths that are inherently easier to integrate. Combining these advanced techniques, high-quality samples can now be generated in tens of sampling steps.

Conceptually, all of these methods reduce the number of sampling steps by increasing the *distance covered by each sampling step*, allowing for fewer, larger steps to reach the data distribution. In this work, we instead propose *Warm-Start Diffusion* (WSD), a method that reduces the *total distance* to be traversed in the first place by moving the initial distribution closer to the data distribution, based on the context information $C$.

Figure 1: Warm-start diffusion targets *strongly conditional* generative tasks, where it yields the largest speed-ups over standard diffusion. In unconditional and weakly conditional tasks WSD works, but achieves no significant acceleration.

## 1.1 SCOPE

This reliance on $C$ makes WSD applicable to any generative task where $C$ is *highly informative*. This domain encompasses many important and computationally expensive domains, such as:

- Image inpainting, super-resolution, noise-removal, and colouration ($C$ = available pixels).
- Video and audio generation ($C$ = previous frames or spectral coefficients).
- Molecule generation ($C$ = molecule properties (Hoogeboom et al., 2022) or graph of atoms (Xu et al., 2022)).
- Weather forecasting ($C$ = current weather) (Kong et al., 2021; Ho et al., 2022; Price et al., 2024).
- Fluid dynamics simulators ($C$ = previous state) (Shu et al., 2023).

Conversely, tasks where $C$ is not a strong constraint, like unconditional diffusion, class-conditional diffusion, or text-to-image generation **are not in scope** for WSD. We visualise the scope in Fig. 1, and consider how our method extends to weakly conditional tasks in Appendix C.

In summary, our contributions include:

- The warm-start diffusion approach, which substantially reduces the computational cost of sampling in strongly conditional diffusion settings.
- A conditional normalisation trick, that makes our method compatible with any standard diffusion framework, and easy to implement.
- A detailed evaluation on image inpainting and weather forecasting tasks demonstrating the method's effectiveness.
- A discussion of the limitations of this method, particularly with regard to unconditional or weakly conditional diffusion domains.

## 2 WARM-START DIFFUSION

Our main contribution is *Warm-Start Diffusion* (WSD) — a method that speeds up sampling in conditional diffusion by moving the noise distribution closer to the data distribution. Instead of drawing the initial noise sample $X_T$ from a standard normal distribution $X_T \sim \mathcal{N}(\mathbf{0}, I)$, WSD uses a small, deterministic *warm-start model* to predict a conditional mean $\hat{\boldsymbol{\mu}}_C$ and marginal standard deviation $\hat{\boldsymbol{\sigma}}_C$ from a given context $C$. Using these moments, a noisy sample can be drawn from the *informed* prior $p(X_T \mid C) = \mathcal{N}(\hat{\boldsymbol{\mu}}_C, \text{diag}(\hat{\boldsymbol{\sigma}}_C^2))$, which we write as $\mathcal{N}(\hat{\boldsymbol{\mu}}_C, \hat{\boldsymbol{\sigma}}_C)$ for brevity. By using this informed prior as the starting point for an entirely separate generative model, we can skip a large number of initial sampling steps. This is illustrated in Fig. 2.

We adopt the DDPM notation, where $t \in [0, T]$ defines a timestep in the sampling process, with $t = 0$ being the final sample from the data distribution and $t = T$ being the initial noise sample.

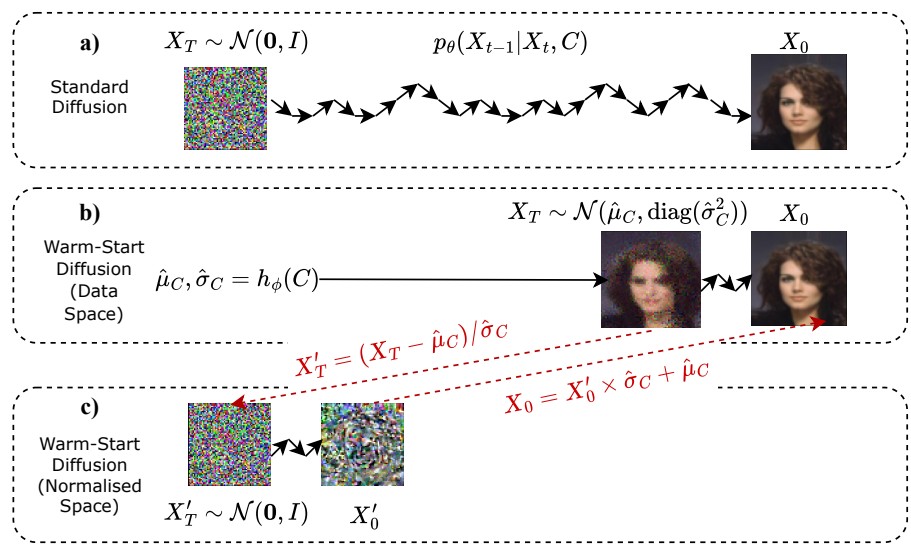

Figure 2: **a)** In standard diffusion, many steps are needed to transform a sample $X_T \sim \mathcal{N}(\mathbf{0}, I)$ to $X_0 \mid C \sim p(X_0 \mid C)$. **b)** Using a warm-start model $h_\phi$, we can draw an initial sample $X_T \mid C \sim \mathcal{N}(\hat{\boldsymbol{\mu}}_C, \mathrm{diag}(\hat{\boldsymbol{\sigma}}_C^2))$ that is already close to the data distribution, allowing us to traverse the gap in fewer steps. **c)** By working in an equivalent sample-normalised space, where $X_T' \sim \mathcal{N}(\mathbf{0}, I)$, a normalised-space sample $X_0' \mid C$ can be drawn using standard diffusion, and is then unnormalised to obtain a sample $X_0 \mid C$ from the data distribution.

## 2.1 GENERATION

The full generative process requires three components:

- Context data $C$ (e.g. fixed pixels in an inpainting task, or the current weather in a weather forecasting task).

- A warm-start model $h_\phi$ that takes the context data $C$ and outputs the first two moments of the conditional data distribution $p(X_0 \mid C)$, i.e. the mean and marginal standard deviation $\hat{\boldsymbol{\mu}}_C$ and $\hat{\boldsymbol{\sigma}}_C$.

- A generative model[1] $p_\theta(X_0 \mid X_T, C, \hat{\boldsymbol{\mu}}_C, \hat{\boldsymbol{\sigma}}_C)$, that generates samples from the conditional data distribution $p(X_0 \mid C)$, given the context data $C$ and a noise sample $X_T \sim \mathcal{N}(\hat{\boldsymbol{\mu}}_C, \hat{\boldsymbol{\sigma}}_C)$.

An explanation of how $h_\phi$ and $p_\theta$ can be trained is given in Section 2.4.

The process to generate a sample $X_0$ from context $C$ is:

$$\hat{\boldsymbol{\mu}}_C, \hat{\boldsymbol{\sigma}}_C = h_\phi(C), \quad X_T \sim \mathcal{N}(\hat{\boldsymbol{\mu}}_C, \hat{\boldsymbol{\sigma}}_C), \quad X_0 \sim p_\theta(X_0 \mid X_T, C, \hat{\boldsymbol{\mu}}_C, \hat{\boldsymbol{\sigma}}_C), \quad (1)$$

which is shown in Figs. 2 and 3.

## 2.2 THE CONDITIONAL NORMALISATION TRICK

Many common diffusion algorithms are derived with the assumption that noise is sampled from a *standard* Gaussian $X_T \sim \mathcal{N}(\mathbf{0}, I)$. To make these diffusion algorithms compatible with WSD, where $X_T \sim \mathcal{N}(\hat{\boldsymbol{\mu}}_C, \hat{\boldsymbol{\sigma}}_C)$, they would potentially need to be re-derived and re-implemented. We sidestep this inconvenience using the conditional normalisation trick.

---

[1] Here, $p_\theta$ is implemented by an iterative solver. When using a deterministic ODE solver, this conditional distribution is a Dirac delta.

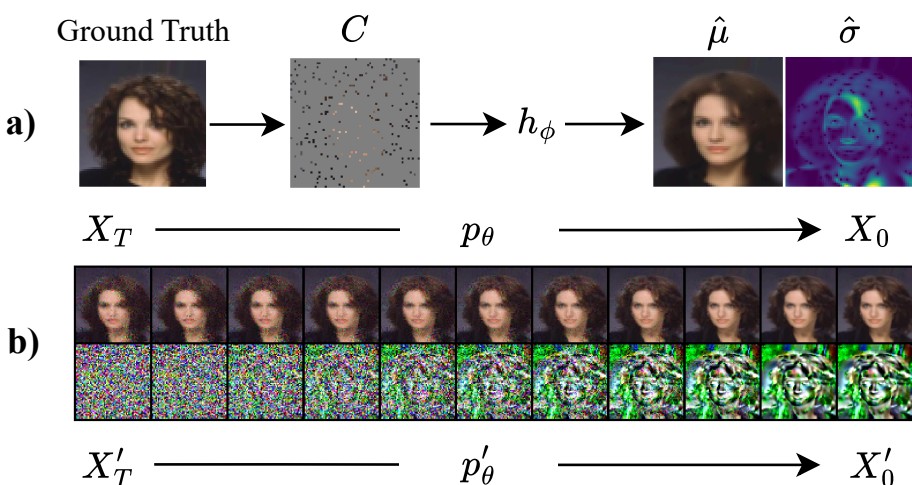

Figure 3: The entire 10-step sampling process for image inpainting. **a)** The context data $C$ is a masked ground truth image with 5% of the pixels visible. The warm-start model $h_\phi$ predicts a conditional mean and marginal standard deviation. **b)** By starting with a sample from $\mathcal{N}(\hat{\boldsymbol{\mu}}_C, \hat{\boldsymbol{\sigma}}_C)$ and applying standard diffusion, a realistic sample $X_0$ is generated. The bottom row shows the same process but in normalised space, where $X_T' \sim \mathcal{N}(\mathbf{0}, I)$.

It is well known that the base distribution $\mathcal{N}(\hat{\boldsymbol{\mu}}_C, \hat{\boldsymbol{\sigma}}_C)$ can be shifted by $\hat{\boldsymbol{\mu}}_C$ and scaled by $\hat{\boldsymbol{\sigma}}_C$ to produce a standard normal $\mathcal{N}(\mathbf{0}, I)$. If we apply the same transformation on a per-instance basis to all steps of the diffusion process $X_t$, the generative model can perform diffusion in an instance-normalised space, $X_t'$:

$$X_t \to X_t' = (X_t - \hat{\boldsymbol{\mu}}_C)/\hat{\boldsymbol{\sigma}}_C. \tag{2}$$

Intuitively, in data space, WSD moves the noise distribution closer to the data distribution. In normalised space, WSD moves *the data distribution closer to the noise distribution*, by *removing* the first two moments from the data distribution. Both approaches are mathematically equivalent, but the latter allows for significantly easier implementation because $X_T' \sim \mathcal{N}(\mathbf{0}, I)$, recovering the standard diffusion assumption. Both are shown in Figs. 2 and 3. Generation in normalised space thus becomes:

$$X_T' \sim \mathcal{N}(\mathbf{0}, I), \quad X_0' \sim p_\theta'(X_0' \mid X_T', C, \hat{\boldsymbol{\mu}}_C, \hat{\boldsymbol{\sigma}}_C), \quad X_0 = X_0' \cdot \hat{\boldsymbol{\sigma}}_C + \hat{\boldsymbol{\mu}}_C. \tag{3}$$

In Sec. 2.4 and Alg. 1, we explain how $p_\theta'$ is trained.

## 2.3 WARMTH BLENDING

We find that WSD in this form significantly improves image quality for low NFE. However, in the high NFE regime, standard flow matching performs better. This is shown as an ablation in Fig. 5 (right, red).

We hypothesise that this underperformance is due to heavy tails in the data distribution, emphasised by conditional normalisation: by removing the first and (marginal) second moments from the data distribution, the heavy tails of the data distribution become exaggerated (as shown in Appendix D). Diffusion models are unable to accurately model heavy-tailed distributions (Pandey et al., 2024), resulting in the observed underperformance.

To overcome this, we introduce the *warmth blending* adjustment: We train the diffusion model on data ranging from entirely *unnormalised* data $w = 0$, corresponding to standard diffusion, to fully normalised data $w = 1$, by modifying $\hat{\boldsymbol{\sigma}}_C$ so that

$$\hat{\boldsymbol{\sigma}}_C^{(\text{norm})} = w \cdot \max(\hat{\boldsymbol{\sigma}}_C, 1 - w) + (1 - w)\mathbf{1} \tag{4}$$

**Algorithm 1** Training Step for $p'_\theta$

1: **Input:** $h_\phi, p'_\theta, \mathcal{D}_{\text{train}}$, optimizer
2: $(C, X_0^{(\text{true})}) \sim \mathcal{D}_{\text{train}}$
3: $(\hat{\boldsymbol{\mu}}_C, \hat{\boldsymbol{\sigma}}_C) \leftarrow h_\phi(C)$
4: $w \sim U[0, 1]$
5: $\boldsymbol{\sigma}_C^{\text{norm}} \leftarrow w \cdot \max(\hat{\boldsymbol{\sigma}}_C, 1 - w) + (1 - w)\mathbf{1}$
6: $X'_0^{(\text{true})} \leftarrow (X_0^{(\text{true})} - \hat{\boldsymbol{\mu}}_C)/\boldsymbol{\sigma}_C^{\text{norm}}$
**\*:** $\mathcal{L} \leftarrow \text{loss}(p'_\theta, C, \hat{\boldsymbol{\mu}}_C, \boldsymbol{\sigma}_C^{\text{norm}}, w, X'_0^{(\text{true})})$
7: $\theta \leftarrow \theta + \text{optimizer}(\nabla_\theta \mathcal{L})$

**Algorithm 2** Warm-start Sampling

1: **Input:** $C, h_\phi, p'_\theta, [w = 1.0]$
2: $(\hat{\boldsymbol{\mu}}_C, \hat{\boldsymbol{\sigma}}_C) \leftarrow h_\phi(C)$
3: $\boldsymbol{\sigma}_C^{\text{norm}} \leftarrow w \cdot \max(\hat{\boldsymbol{\sigma}}_C, 1 - w) + (1 - w)\mathbf{1}$
4: $X'_T \sim \mathcal{N}(0, 1)$
**\*:** $X'_0 \sim p'_\theta(X'_0 \mid X'_T, C, \hat{\boldsymbol{\mu}}_C, \boldsymbol{\sigma}_C^{\text{norm}}, w)$
5: $X_0 \leftarrow X'_0 \cdot \boldsymbol{\sigma}_C^{\text{norm}} + \hat{\boldsymbol{\mu}}_C$
6: **return** $X_0$

**\***Note that we do not prescribe how to sample from $p'_\theta$, or how its loss is calculated, as WSD is agnostic to the implementation of the generative model.

is used for (un)normalisation. We also pass $w$ to $p'_\theta$ as an additional scalar input. During training (Alg. 1), $w$ is randomly sampled $w \sim \text{U}[0, 1]$. During inference (Alg. 2), $w$ is a hyperparameter, which we simply set to 1 for all experiments[2].

This training curriculum blends the well-modelled unnormalised space $w = 0$ with the heavy-tailed normalised space $w = 1$, and forces the model to learn how to continuously transform the former into the latter. Empirically, we find that with warmth blending, **WSD outperforms standard diffusion for all NFE**. Appendix D explains this mechanism in more detail.

## 2.4 TRAINING

The goal of training is to learn the warm-start model $h_\phi$ and the normalised-space generative model $p'_\theta$ required for sampling. These two models can be trained simultaneously (and even in an end-to-end manner, with caveats explained in Appendix E), and implemented, stored, and used as one singular generative model, resulting in very little added engineering complexity.

However, training can also be split up into two sequential phases, where we first train $h_\phi$ and then $p'_\theta$. This modular approach has the following benefits:

- $h_\phi$ may be useful as a deterministic model even without $p'_\theta$. For instance, in weather forecasting, both deterministic models and generative models are useful in different contexts (Couairon et al., 2024).

- Any existing Gaussian regression model can be used as $h_\phi$ without a need for retraining.

- Once $h_\phi$ is trained, its per-sample outputs can be cached, saving memory and compute when training $p'_\theta$.

**Training the Warm-Start Model**  The goal of the warm-start model is to predict the first two moments of the conditional data distribution $p(X_0 \mid C)$. We do this by training a probabilistic regression model $h_\phi$ with parameters $\phi$ using Gaussian negative log-likelihood loss, inspired by conditional neural processes (Garnelo et al., 2018a;b):

$$\mathcal{L}_\phi = -\log p_\phi(X \mid C) = -\log \mathcal{N}(X \mid \hat{\boldsymbol{\mu}}_C^{(\phi)}, \hat{\boldsymbol{\sigma}}_C^{(\phi)}). \tag{5}$$

Once $h_\phi$ is trained, we freeze its weights.

**Training the Generative Model**  Training the normalised-space generative model $p'_\theta$ is straightforward: the only difference to a standard training step is that training samples are instance-normalised based on the outputs of $h_\phi(C)$ (see Sec. 2.2). This works for any off-the-shelf generative model, which is why WSD is model-agnostic. This is shown, for a single training sample, in Alg. 1.

---

[2]We find that using slightly smaller values of $w = 0.8$ in the high NFE regime yields very slightly better FID scores, but find these gains to be visually imperceptible and not worth the additional complexity of adapting $w$.

## 2.5 WSD AS A GENERALISATION OF STANDARD DIFFUSION

We note that although the diagonal Gaussian $\mathcal{N}(\hat{\boldsymbol{\mu}}_C, \hat{\boldsymbol{\sigma}}_C)$ is a simplified approximation of the true conditional distribution, *standard diffusion makes this same approximation*, with the *additional restriction* that the moments are fixed to $\mathbf{0}$ and $\mathbf{1}$. Standard diffusion can thus be viewed as a special case of WSD. This implies that a warm-start model trained to minimise the NLL (Eq. 5) provides a starting point that is *at least as good* as the uninformed prior used in standard diffusion.

## 3  RELATED WORK

Other generative methods that are fast at inference time exist, but each has its own shortcomings. Generative adversarial networks (Goodfellow et al., 2020) can generate images in a single forward pass but are difficult to train and can suffer from mode collapse. Consistency models (Song et al., 2023) are modern alternatives that learn to map any point on the diffusion trajectory directly to the data distribution. While powerful when trained, they require a complex two-stage training process involving the distillation of a pre-trained diffusion model. This distillation process can be brittle and computationally expensive, relying on careful scheduling and large synthetic datasets.

Standard diffusion assumes a trajectory from pure noise $\mathcal{N}(\mathbf{0}, I)$ to data. However, recent theoretical frameworks such as Schrödinger Bridges (Liu et al., 2023) and Stochastic Interpolants (Albergo et al., 2023) generalise this to trajectories connecting arbitrary distributions, like a corrupted image and a clean image (Liu et al., 2023). WSD can be viewed as a tractable and lightweight instantiation of this framework, bridging between a learned diagonal Gaussian $\mathcal{N}(\hat{\boldsymbol{\mu}}_C, \hat{\boldsymbol{\sigma}}_C)$ and the data.

Some recent methods attempt to accelerate diffusion by modifying the sampling trajectory. Leapfrog diffusion models (Mao et al., 2023) introduce a trainable initializer to estimate a denoised distribution at an intermediate timestep, allowing the model to skip some denoising steps. Similarly, shortcut models (Frans et al., 2024) condition the network on a desired step size, allowing it to learn velocity fields at different granularities to take larger steps. In contrast to WSD, these methods attempt to skip steps within the diffusion process, whereas WSD adjusts its starting point. WSD is orthogonal to these methods and could be combined with them to further accelerate generation.

In the domain of weather forecasting, single-step generative models relying on the Continuous Ranked Probability Score (CRPS) have shown recent success (Lang et al., 2024; Alet et al., 2025), but this method is domain-specific and potential shortcomings are not yet fully understood[3].

## 4  EXPERIMENTAL SETUP

Across our experiments, we use the Meta Research implementation of flow matching (Lipman et al., 2024; 2022) as our baseline, but warm-start models can be combined with any diffusion-based algorithm. We combine this model with the state-of-the-art V3 DPM-Solver (Zheng et al., 2023). To make DPM Solver compatible with the flow-matching formalism, we use the equivalence to noise-based diffusion outlined in Gao et al. (2024). To the best of our knowledge, this is the first time flow matching and DPM Solver are combined, creating a very strong sample-efficient baseline. As flow matching and diffusion can be shown to be different formulations of the same principle (Gao et al., 2024; Patel et al., 2024), we use both terms interchangeably.

To keep comparisons fair, we use the same architecture for both the baseline and our (warm-start) generative models. Additionally, our warm-start model is kept significantly smaller than the generative model, so that one forward pass takes around 1/10th of the time of the generative model. For brevity, we do not include this faster forward pass in our NFE numbers (i.e. we write NFE=10 instead of NFE=1 fast + 10 slow). For more experiment details, including the model architecture choices, see Appendix B.

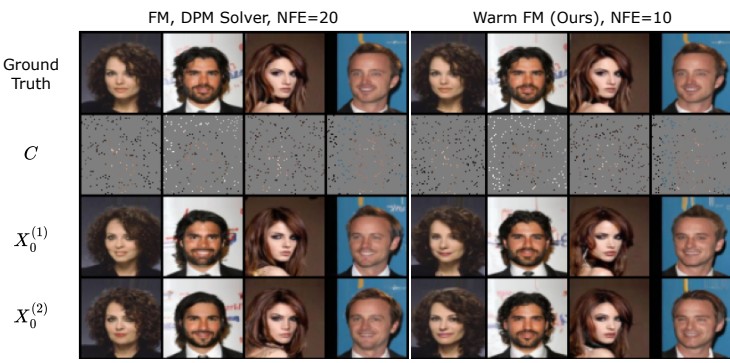

Figure 4: Samples $X_0^{(i)}$ generated by standard Flow Matching (NFE=20) and our method (NFE=10).

## 5 IMAGE INPAINTING

In this task, we select a random image from the relevant dataset, and randomly mask out 95% of the pixels in the image (90% for CIFAR10 due to the lower resolution). This masked image (as well as the mask itself) is then used as the context data $C$, as shown in Fig. 4.

The models' task is to generate a sample $X_0$ that matches the masked image, i.e. fills in the missing pixels, while remaining consistent with the unmasked pixels. The entire sampling process is shown in Figure 3.

### 5.1 RESULTS

We evaluate our method on the 64x64 CelebA (Liu et al., 2015) and the 32x32 CIFAR10 (Krizhevsky, 2009) datasets. In both settings, we discard any labels and supplementary information, and only use the masked images (as well as the mask itself) as context data $C$.

As shown in Fig. 4, our method generates realistic samples that are consistent with the unmasked pixels, despite only using NFE=10. These samples are competitive with traditional flow matching using the DPM solver and NFE=20. Additional samples (including for CIFAR10) can be found in Appendix G.

For quantitative evaluation of perceptual quality, we use the FID (Fréchet inception distance) (Heusel et al., 2017), computed over 50,000 samples, each evaluated for NFEs between 2 and 100 (Fig. 5). Clearly, in the low NFE regime, our method substantially outperforms standard flow matching, able to generate perceptually realistic images using NFE= $4 - 6$, and saturating performance in 12. Individual samples at different NFE are shown in Appendix G (Figs. 11, 12). We also find that our method slightly outperforms the baseline in the high NFE regime. We believe this to be mainly due to the mean subtraction making the modelling task easier, as explained in the mean-only ablation (Sec. 5.2).

We extensively experiment with various general-purpose and diffusion-specific ODE solvers and integration time discretisations and plot only the best-performing combination at each NFE value. This is generally the midpoint solver using uniform time discretisation for low NFE values (NFE $\leq 5 - 10$), and the 3rd order DPM Solver using the log-signal-to-noise-ratio time discretisation for NFE $> 5 - 10$. See Appendix B.1 for more details.

### 5.2 ABLATIONS

All ablations are performed against the CIFAR10 dataset. We do not extend these ablations to other datasets due to computational constraints.

---

[3]For instance, as the CRPS only considers marginal distributions, the loss does not inherently guarantee realistic joint distributions.

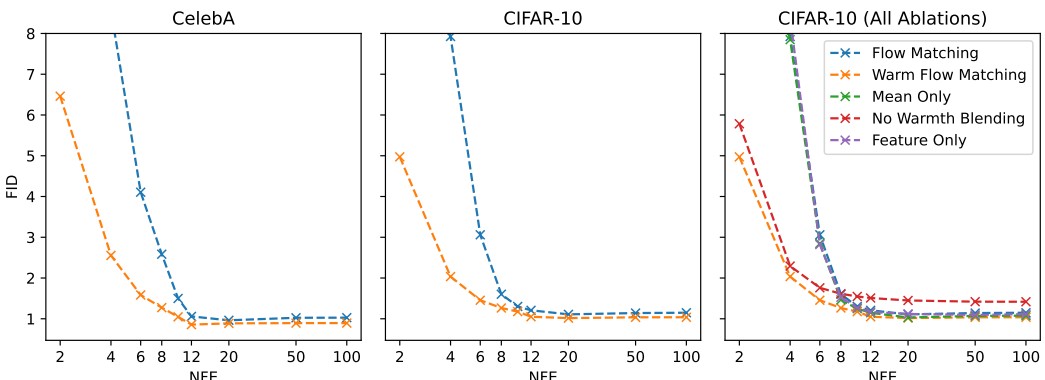

Figure 5: Warm-start flow matching substantially outperforms its standard counterpart in the low NFE regime, allowing high-quality samples to be generated in 4-6 function evaluations, and saturating performance in 12.

**No warmth blending**  Here, we retrain a model without the warmth blending and multi-task training described in Sec. 2.3. This is shown in Fig. 5 (right, red). Clearly, while the model is still far more NFE-efficient than standard flow matching, it underperforms the blended-warmth model (orange) at all NFE.

**Mean-only**  Here, we only use the predicted mean $\hat{\boldsymbol{\mu}}_C$ for normalisation (equivalent to setting $\hat{\boldsymbol{\sigma}}_C = \mathbf{1}$). This is equivalent to training a deterministic (R)MSE model (outputting $\hat{\boldsymbol{\mu}}_C$) as the shortcut model, and performing diffusion against the residuals. This has shown success in weather forecasting models (Couairon et al., 2024; Mardani et al., 2025). Performance is visualised in Fig. 5 (right, green). Compared to normal flow matching, performing diffusion in the residual space improves performance slightly, indicating that this is where our method's *high-NFE gains* come from. In the low-NFE regime, the mean-only normalisation performs as poorly as standard flow matching. This also shows that the efficiency gains demonstrated using WSD *heavily depend on* $\hat{\boldsymbol{\sigma}}_C$. Assuming uniform noise (i.e. $\hat{\boldsymbol{\sigma}} = 1$) ignores which regions of the image are well-constrained by context, applying too much noise to most areas, which must then be iteratively removed by the generative model.

**Features only**  It could be the case that the increased efficiency comes not from moving $X_T$ closer to $X_0$, but instead from the fact that the generative model $p_\theta$ has access to $\hat{\boldsymbol{\mu}}_C, \hat{\boldsymbol{\sigma}}_C$ as inputs. In this case, our method works effectively as a form of feature engineering. We test this by not applying the normalisation, but still providing $\hat{\boldsymbol{\mu}}_C, \hat{\boldsymbol{\sigma}}_C$ as inputs to the generative model. As shown in Fig. 5 (right, purple), this yields no significant improvement over the standard flow matching baseline, demonstrating that the observed benefits come from the warm-start approach itself, not the additional inputs.

## 6  ERA5 WIND FORECASTING

In ML-based weather forecasting, the goal is to predict the future weather given the current weather. These systems typically operate on a fixed time interval (e.g. 6 hours). To produce predictions on longer time horizons, the model is applied autoregressively. As the model is trained on *real* weather samples, but deployed autoregressively (using *its own* predictions as inputs), model outputs must be *realistic* weather samples. Otherwise, the model falls increasingly out of distribution when rolled out in time.

Existing diffusion-based generative models such as GenCast (Price et al., 2024) have shown good results, but are expensive to run. For instance, a single 15-day forecast with 50 ensemble members at NFE=39 per sample (as performed by Price et al. (2024)) requires 58,500 forward passes (see Appendix F), needing $\sim$ 7 hours on a single Cloud TPUv5 device (Price et al., 2024). As shown

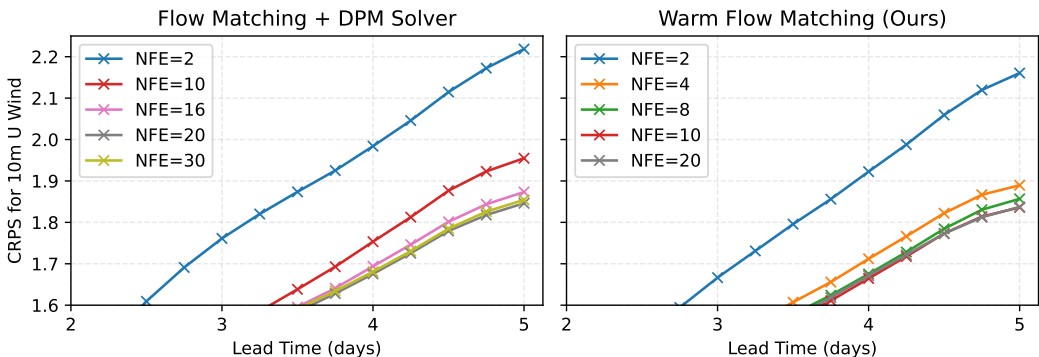

Figure 6: Continuous Ranked Probability Score (CRPS) computed over an ensemble of 50 forecast trajectories. With conventional flow matching and DPM Solver (left), the CRPS performance saturates for NFE above $\sim 20$. Using warm-start flow matching (right), performance saturates after NFE=10. The *saturated* performance of both methods is very similar.

in Fig. 7, our method requires only NFE $\approx 10$ per AR Step, reducing compute requirements by $\sim 75\%$.

We emphasise that our goal is not to achieve state-of-the-art forecasting results, but rather to demonstrate that our method can generate realistic weather samples in a fraction of the sampling steps used by current approaches. To do this, we use a lightweight convolutional U-Net (Ronneberger et al., 2015) architecture, and restrict ourselves to only modelling the $u$ and $v$ components of wind 10m above the ground. We also limit ourselves to a spatial resolution of $1.5°$ (i.e. 240x121 grid points), as provided by the re-gridded ERA5 reanalysis dataset (Hersbach et al., 2020). Our model uses an internal temporal resolution of 6 hours, and is given a snapshot of the current wind fields and the wind fields 6 hours prior as context data $C$.

## 6.1 RESULTS

In the absence of a perceptual accuracy metric like the FID for generated images, we evaluate our models using two commonly used metrics:

1. Fig. 6 shows the Continuous Ranked Probability Score (CRPS) over a 5-day autoregressive forecast using 50 ensemble members. The CRPS is a proper scoring rule which can be considered as a probabilistic generalisation of the mean absolute error.

2. Fig. 7 shows the power spectrum ratio $\eta(\lambda)$. It compares the power of different wavelengths $\lambda$ present in generated samples to the ground truth power. Good samples have $\eta(\lambda) \approx 1 \, \forall \, \lambda$.

In both metrics, standard flow matching (with DPM Solver) shows improvements up to NFE $\approx 20$, whereas WSD saturates performance for NFE above $\approx 10$. Appendix G (Fig. 13) visualises forecast trajectories sampled using WSD as well as the ground truth, showing that the warm-start model is capable of generating plausible, yet diverse forecasts.

## 7 CONCLUSION

In this work, we introduced warm-start models, a widely applicable, easily implemented, and effective method for reducing the NFE required in conditional generative modelling, without sacrificing quality in the high-NFE regime. By using a simple, deterministic network to predict the initial moments of the conditional data distribution, we effectively reduce the distance the generative process must traverse. This approach works with any standard generative model, is orthogonal to and synergistic with existing efficient samplers, is simple to implement, and is computationally cheap at $\sim 10\%$ of the total training cost. On benchmark tasks like image inpainting and weather forecasting, our approach can generate realistic samples in 4-6 function evaluations, and saturates performance in 10-12, demonstrating a substantial leap in sampling efficiency.

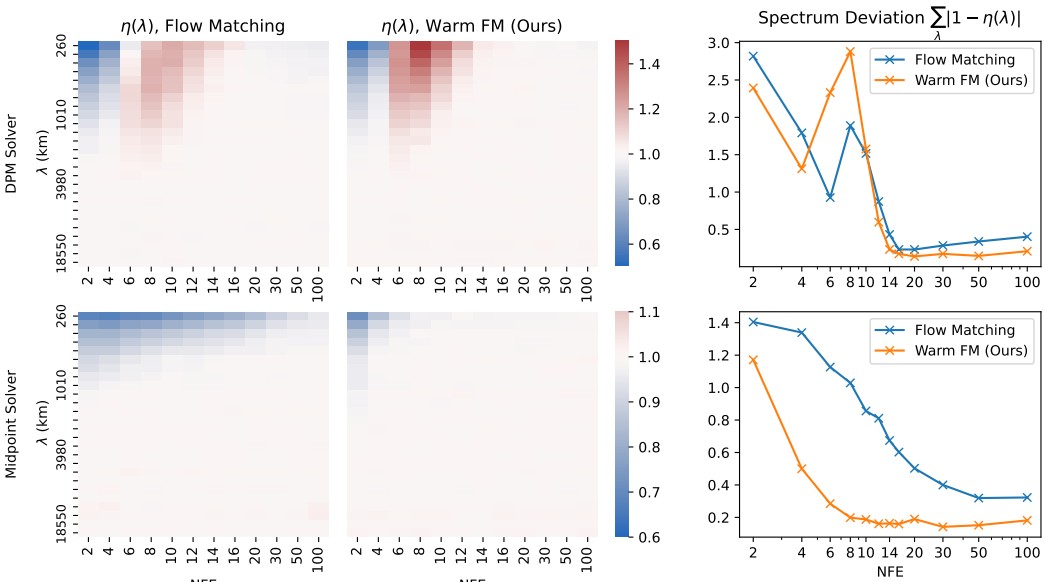

Figure 7: **Left**: The power spectrum ratio, $\eta(\lambda)$, compares the presence of certain wavelengths in the model's predictions to the ground truth: $\eta(\lambda) < 1$ (blue) $\implies$ $\lambda$ is under represented, $\eta(\lambda) > 1$ (red) $\implies$ $\lambda$ is overrepresented. For low NFE, predictions are blurry. For higher NFE, the generated samples' power spectra align with the ground truth. **Right**: By summing the absolute deviations from the ground truth power spectrum $\sum_\lambda |1 - \eta(\lambda)|$, we can summarise the power spectrum deviation into a single number at each NFE. **Top row**: Using DPM Solver, both standard and warm-start flow matching reach their terminal state after $14 - 20$ NFE. **Bottom row**: Using the midpoint solver, warm-start flow matching (orange) becomes significantly more efficient than conventional flow matching, needing only $\sim$ NFE=10 to saturate its performance.

**Limitations** The primary limitation of this method lies in the warm-start model's assumption of an uncorrelated Gaussian posterior. This makes it highly effective for tasks with strong conditioning information that lead to a largely unimodal conditional distribution, such as inpainting or weather forecasting. Conversely, its utility is diminished in highly multimodal settings like text-to-image synthesis, where a single Gaussian is an insufficient prior. Further work is needed to investigate how WSD performs on more multimodal tasks with weaker conditioning information (e.g. inpainting with fewer pixels or weather forecasting over longer time intervals). A second limitation is that a separate warm-start model needs to be trained for each experiment and dataset. It may be possible[4] to train a single general-purpose warm-start model (trained e.g. on Imagenet Deng et al. (2009)) that can be used for any image-related tasks.

**Future work** WSD can be made even more efficient and flexible. Predicting a conditional low-rank correlation matrix, instead of only marginal standard deviations, could accelerate the method. Additional speed-ups may come from adapting efficient sampling tricks, like EDM's custom time discretisation (Karras et al., 2022) or ODE solvers such as DPM-Solver Lu et al. (2022; 2025); Zheng et al. (2023), from standard diffusion to WSD. Finally, WSD opens up the possibility of inference-time compute scaling: by using the uncertainty estimate from the warm-start model to allocate the number of sampling steps (using more for highly uncertain predictions and fewer for confident ones), compute can be dynamically allocated based on need.

These advancements, building upon an already simple and effective framework, have the potential to make WSD an even more efficient and flexible tool for conditional generation.

---

[4]In fact, we mistakenly initially used a CIFAR10-trained warm-start model for WSD on CelebA. We found only a small performance loss even though the two datasets are substantially different.

## REPRODUCIBILITY STATEMENT

We make our method reproducible by outlining the method in Sec. 2, providing the broad experimental setup in Sec. 4, providing more details in Appendix B, and also providing the anonymised source code for review. After anonymous peer review, we will make the source code available on GitHub.

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

## A    LLM Declaration

We used LLMs to assist with writing code and iterating on the language in the final paper.

## B    Experimental Details

**Datasets**    All datasets are normalised. For images, we normalise values to lie between [-1, 1]. For the weather forecasting task, we apply a per-variable normalisation to ensure zero-mean and unit variance.

**Warm-start model**    We parameterise $h_\phi$ as a lightweight U-Net (Ronneberger et al., 2015) with [64, 128, 256] channels per block and 2 layers per block. We use attention in the second and third blocks. For the weather forecasting task, we instead use [128, 256, 512] channels, but no attention (as the resolution is much higher, and attention would become computationally expensive). We train the warm-start model until convergence ( $\approx$ 2 million steps) at a batch size of 32 using AdamW at a constant learning rate of 1e-4 (and using default weight decay and betas). We clip the predicted standard deviation at 0.01 to stabilise training and avoid numerical instability when performing normalisation. For the inpainting tasks, we train the model over a range of inpainting tasks, ranging from 3% of pixels to 10% of pixels for CelebA, and 5% of pixels to 20% of pixels for CIFAR10.

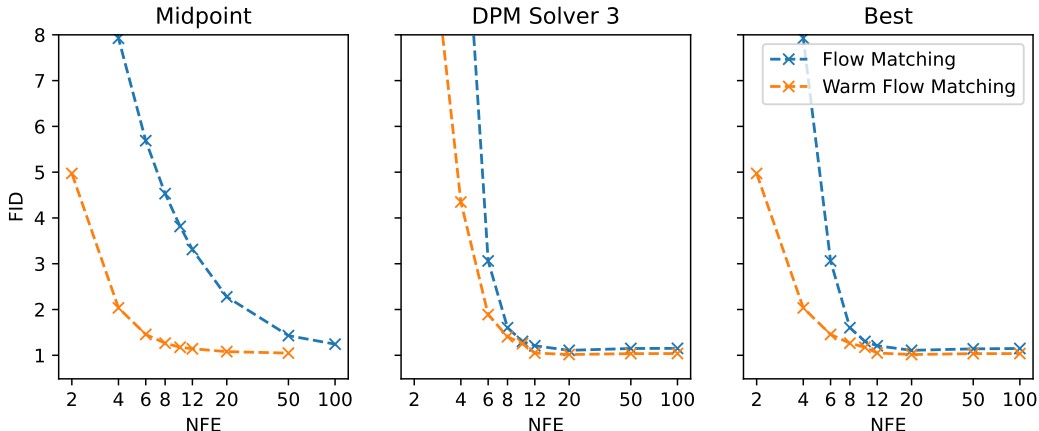

Figure 8: On CIFAR10, warm-start diffusion substantially outperforms its standard "cold" counterpart in the low NFE regime, allowing high-quality samples to be generated in 6 function evaluations, and saturating performance in 12. The performance gap is very pronounced for the simpler midpoint solver (left). Using DPM Solver makes standard flow matching more competitive (middle), but when using the best solver at each NFE, the performance gain

**Generative model** We choose to follow Lipman et al. (2024) in the model architecture and training procedure for $p'_\theta$. In particular, we use the same U-Net architecture, and train it using the AdamW optimiser (Loshchilov & Hutter, 2019) with a constant learning rate of 1e-4, and with $\beta_1 = 0.9, \beta_2 = 0.95$. We train using an effective batch size of 512 until convergence ($\approx 1.5$ million steps). We condition the model on the diffusion timestep $t$ and the warmth $w$ by computing embeddings and using them to shift and scale features after normalisation. We use exponential moving average (EMA) weight smoothing with a rate of 0.999. We clip gradients with norms above 3.0. For the weather forecasting experiment, we use a batch size of 4, also training until convergence.

For full details, we refer to the provided source code, and particularly the configuration files.

## B.1 Best Solvers

When comparing results (e.g. in Fig. 5), we evaluate each data point using a combination of ODE solvers and time discretisations. We find that in the very low NFE regime ($\leq 5$ for standard diffusion, $\leq 10$ for warm start diffusion), the best results are achieved using the midpoint ODE solver using a uniform time discretisation. For higher NFE, we find that the 3rd order DPM Solver using a log signal-to-noise ratio time discretisation achieves the best results. For very high NFE ($> 50$), we sometimes find that performance slightly degrades using DPM Solvers.

We tested an extensive selection of ODE solvers and time discretisations. Specifically, we test all fixed step solvers available in the torchdiffeq library (Chen, 2018), and the following time discretisation schemes:

- Uniform in time

- Quadratic in time

- Log signal-to-noise ratio

- The EDM discretisation proposed in Karras et al. (2022).

We find that these choices have a large impact on sample efficiency, and we also find that warm-start diffusion is more robust to suboptimal choices than standard diffusion. A selection of results produced by different solvers is shown in Fig. 8.

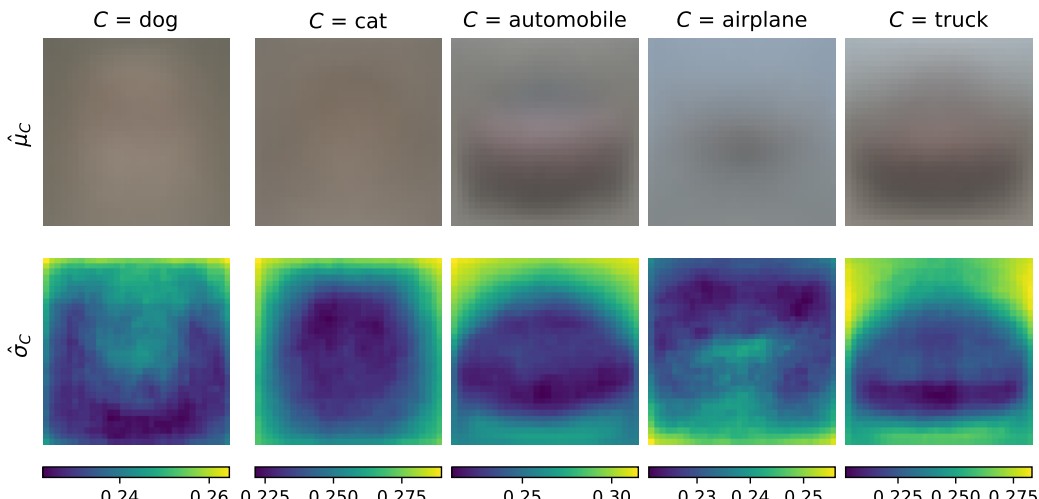

Figure 9: The mean and (greyscale) standard deviation of CIFAR-10 images by class $C$. Note that the means are extremely blurry and the standard deviations are wide (and relatively uniform, as shown by the scales on the colour bars). WSD is unlikely to bring many benefits in this regime.

## C    WEAKLY-CONDITIONAL TASKS

Weakly-conditional tasks, like text-to-image generation, class-conditional generation, or even unconditional generation, are not in scope for WSD. However, a simple theoretical argument shows how WSD would behave in these tasks, and demonstrates that it gracefully approaches standard diffusion in this regime.

The reason why normal WSD needs a warm-start model is that the conditional mean and standard deviation $\hat{\mu}(C), \hat{\sigma}(C)$ are non-trivial functions that must be learned from the training dataset, because each piece of context information (set of visible pixels in the inpainting task, current weather in the wind forecasting task) only appears *once* within the training set. For weakly conditional or unconditional tasks, this is not the case, and the conditional mean and conditional standard deviation can be simply estimated directly from the training data.

Consider unconditional diffusion: the $\hat{\mu}, \hat{\sigma}$ that minimise the Gaussian log-likelihood (Eq. 5) can be computed as the per-pixel sample mean and sample standard deviation computed over the entire training set.

Similarly, in class-conditional diffusion, where $C$ is a class, the optimal $\hat{\mu}_C, \hat{\sigma}_C$ are the sample means/standard deviations computed over the class $C$ in question. For example, consider CIFAR-10: the $\hat{\mu}_{\mathrm{dog}}$ that minimises the Gaussian log-likelihood is simply the mean image computed over all training samples of class "dog", and similarly for $\hat{\sigma}_{\mathrm{dog}}$. This also demonstrates why WSD is unlikely to be very effective in this regime: $\hat{\mu}_{\mathrm{dog}}$ is an extremely blurry mean, with a very wide standard deviation at every point, effectively approaching standard diffusion where the prior is $\mathcal{N}(\mathbf{0}, I)$. We show this in Fig. 9.

## D    EFFECTS OF CONDITIONAL NORMALISATION ON THE DISTRIBUTION OF PIXELS

As shown in Fig. 10 (left), the CelebA pixel values are not normally distributed. However, in standard diffusion, the prior $\mathcal{N}(0, 1)$ covers the data well (because the standard deviation 1 is large enough to cover any pixel value in $[-1, 1]$). Removing the first two moments from the data distribution via conditional normalisation (Sec. 2.2) yields a distribution with heavy tails (Fig. 10 right). Some of the data lies in regions with effectively zero probability mass in the $\mathcal{N}(0, 1)$ prior.

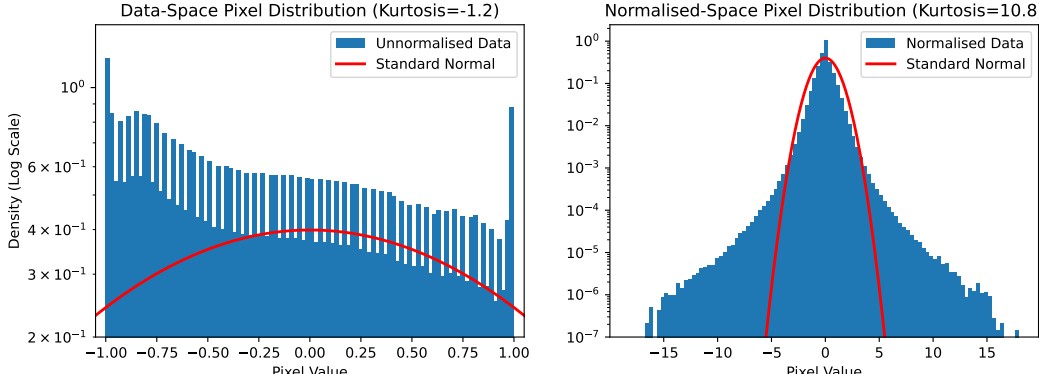

Figure 10: Distribution of pixel values of the CelebA dataset. **Left:** without conditional normalisation, pixels lie in $[-1, 1]$. The data is not Gaussian, but the standard prior $\mathcal{N}(0, 1)$ covers the entire data range. **Right:** Using the warm-start model to remove the first two moments from the data distribution results in heavy tails. The prior $\mathcal{N}(0, 1)$ assigns (essentially) no mass to the outliers.

**The Cause of Heavy Tails**   To better understand the cause of the heavy tails in the conditionally-normalised data, consider the following example: the warm-start model attempts to predict $\hat{\mu}_C, \hat{\sigma}_C$ for a pixel $X^*$ that is part of a white wall in the background of an image. Surrounding pixel values are white ($X = 1$). Further assume the following ground truth distribution: 99% of the time, the target pixel is also white $X^* = 1$, but 1% of the time, there is a black piece of dirt at the target pixel $X^* = -1$. A **perfect warm-start model** would predict the true mean

$$\mu_C = 0.99 \times 1 + 0.01 \times -1 = 0.98,$$

and true standard deviation

$$\sigma_C = \sqrt{\mathbb{E}[x^2] - \mu_C^2} \approx 0.2.$$

Using $\mu_C$ and $\sigma_C$ for conditional normalisation would therefore transform this pixel to a value of $\approx -10$ standard deviations from the mean, corresponding to effectively zero probability.

**Why Warmth Blending Helps**   In *standard diffusion*, the prior covers the entire dataset (see Fig. 10, left). The generative model learns a transport map from the noise $X_T$ to the image $X_0$. For the black speck example, the model simply learns to map the tail of the noise distribution (e.g., the bottom 1%, $x_T \lesssim -2.3$) to the black pixel value $x_0 = -1$, and the bulk of the noise ($x_T \gtrsim -2.3$) to the white pixel $x_0 = +1$. The required transport distances are small and comparable.

In WSD without warmth blending, the conditionally normalised space assigns (effectively) zero probability to the outliers. For the generative model to produce the black speck (at $-10\sigma$), it must learn that for a specific 1% of noise samples, the transport velocity must be *massive and negative* ($v \approx -10$), drastically *increasing the noise* to reach the outlier. However, for the other 99% of noise samples, the required velocity is small, and acts to *reduce* the noise (magnitude $\approx +0.1$), allowing the process to reach the white wall value.

The model is required to learn a very sharp change in behaviour for two very similar inputs, covering a region of the velocity field only seen very rarely during training because of the narrow prior. The model (trained with MSE) effectively ignores the outliers, and learns to always generate the much more common mode corresponding to the white pixel.

Warmth blending bridges this gap: The generative model is forced to learn a continuous representation of "heavy-tailed-ness", as captured by $w$. It can learn to denoise specks of dirt in the small $w$ setting, where the prior covers these outliers, while simultaneously learning how the transport velocity (and its magnitude) change as a function of $w$.

We note that this is a *hypothesis* of why warmth-blending works, and that further work is necessary to confirm this mechanism.

## E END-TO-END TRAINING

As the generative loss is a differentiable function of both the warm-start model's parameters and the generative model's parameters, end-to-end training may appear as a reasonable option to improve performance. In practice, this leads to a reduction in *loss*, but the method collapses entirely because the two models "collude".

Because the generative model $p_\theta$ minimises a denoising objective, the warm-start model $h_\phi$ is incentivized to output parameters $\hat{\boldsymbol{\mu}}_C, \hat{\boldsymbol{\sigma}}_C$ that make the noised data $X_t$ trivial to denoise, rather than accurately modelling the data distribution. For example:

- For noise-predicting formulations, $\hat{\boldsymbol{\sigma}}_C$ can be set to a very large value compared to the data, making it trivial for the generative model to predict the noise.
- For formulations targeting the clean image, $\hat{\boldsymbol{\mu}}_C$ can be set to 0, and $\hat{\boldsymbol{\sigma}}_C$ to a small number. Then, intermediate steps $X_t \approx X_0 \times (1 - \frac{t}{T})$ contain almost no noise, making it trivial to predict $X_0$ (for all $t \neq T$) during training.

Of course, this does not result in a competent generative model.

**End-to-end Training with Partially Detached Gradients**   This "collusion" can be circumvented by detaching the gradients from the predicted standard deviation $\hat{\boldsymbol{\sigma}}_C$, (after line 3 of Alg. 1). The two models $h_\phi, p'_\theta$ can then be trained jointly to minimise the generative loss, without trivialising the end-to-end process. Initial experiments suggest that this can improve performance further, but more work is required to determine the best way to perform end-to-end training, particularly with regards to simultaneously training $\hat{\boldsymbol{\sigma}}_C$.

## F NFE CALCULATION WEATHER FORECASTING

A 15-day forecast with 50 ensemble members at NFE=39 per sample (as performed by Price et al. (2024)) requires:

$$50 \text{ Ens. Members} \times \frac{15 \text{ Days}}{\text{Ens. Member}} \times \frac{2 \text{ AR Steps}}{\text{Day}} \times \frac{39 \text{ Fwd. Passes}}{\text{AR Step}} = 58,500 \text{ Fwd. Passes.} \quad (6)$$

## G ADDITIONAL SAMPLES

We compare warm-start diffusion to standard diffusion qualitatively at different NFE in Figs. 11 (CIFAR10) and 12 (CelebA), showing that details appear for lower NFE values when using WSD.

In Fig. 13, we show a 3-member ensemble of 5-day wind forecasting trajectories. In Figs. 14 and 15, we provide additional samples for CIFAR10 and CelebA inpainting respectively.

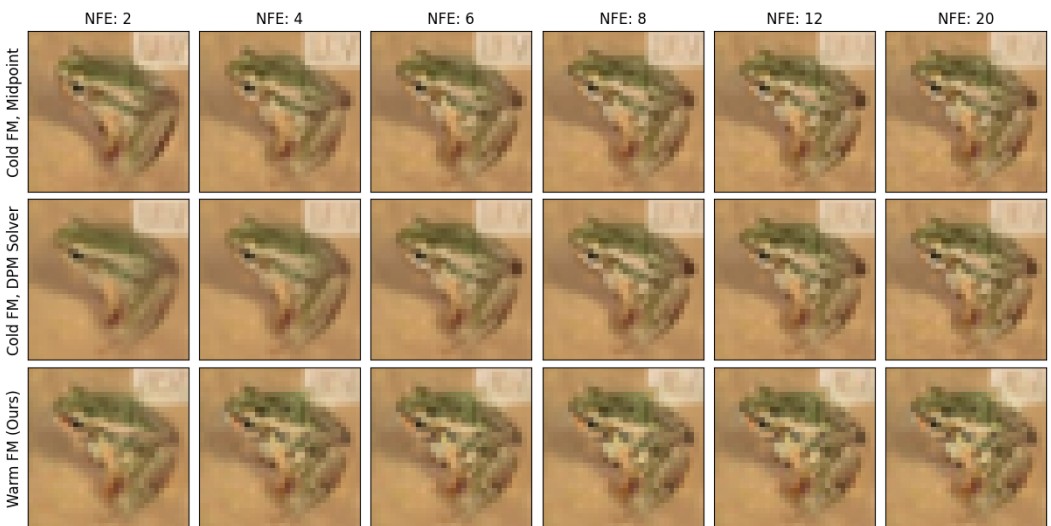

Figure 11: Evaluating samples drawn from the same context and same random noise at different NFE. While standard diffusion produces blurry samples for NFE=2-4, warm diffusion is already able to include high-frequency details. For warm diffusion, past NFE $\sim 4 - 6$, the samples do not visibly change. For standard diffusion, even when using DPM Solver, additional details in the frog's skin texture appear for NFE up to $\sim 12 - 20$.

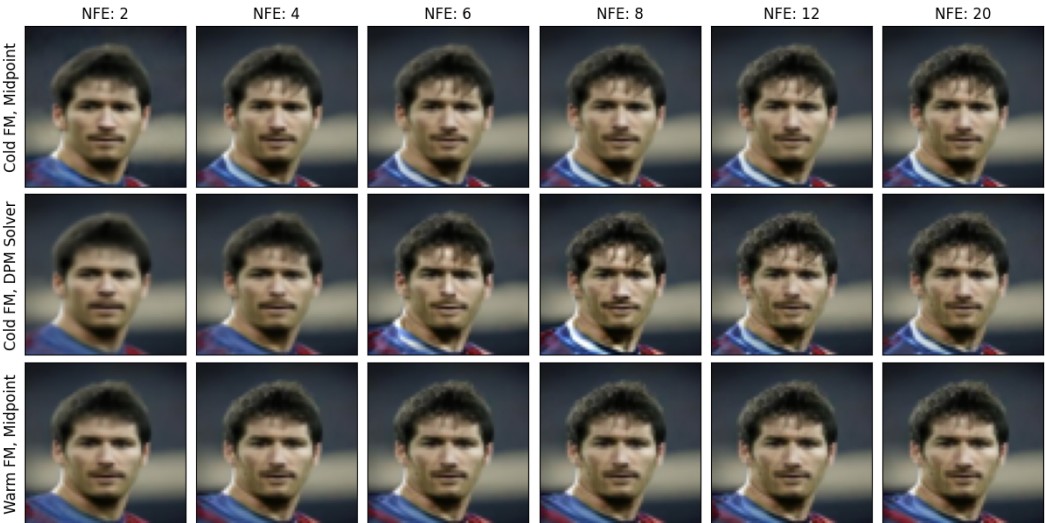

Figure 12: Like Fig. 11 but for the CelebA dataset.

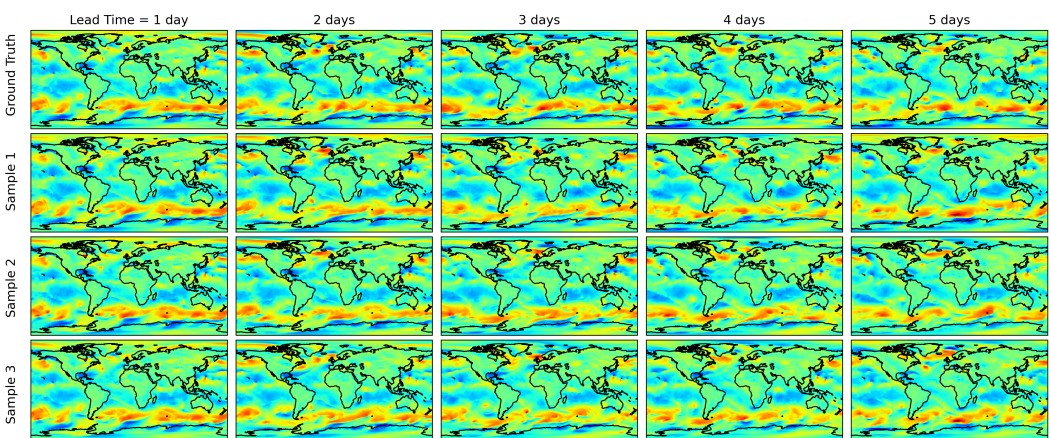

Figure 13: Autoregressive forecast trajectories for the U-component of wind at 10m, generated using NFE=10. **Top row**: Ground truth ERA5 data. **Bottom three rows**: Four independent forecast samples generated by our method (NFE=11 per 6-hour step), starting from the same initial conditions. The forecasts remain plausible and diverge from each other, demonstrating the model's ability to produce a probabilistic ensemble.

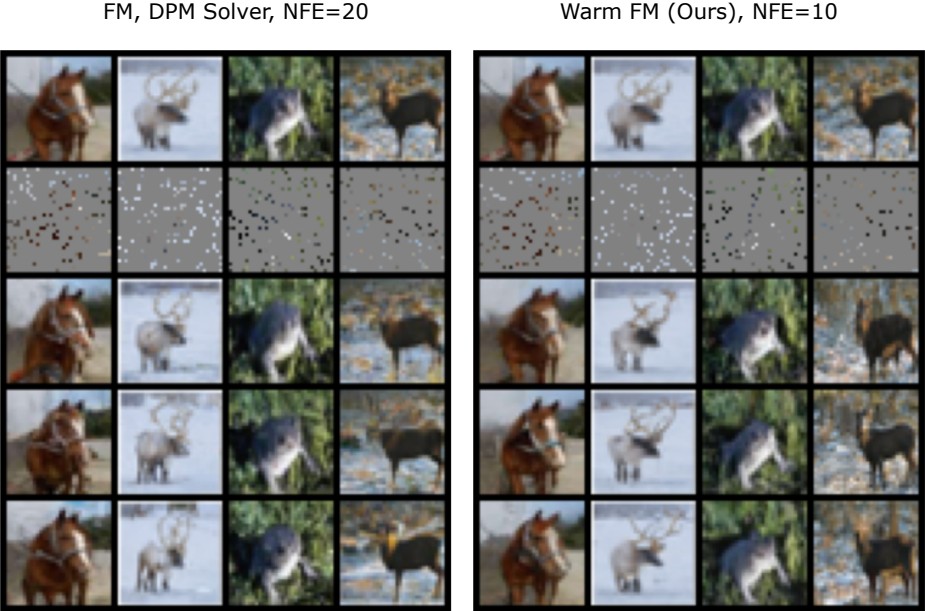

Figure 14: Like Fig. 4 but for CIFAR10.

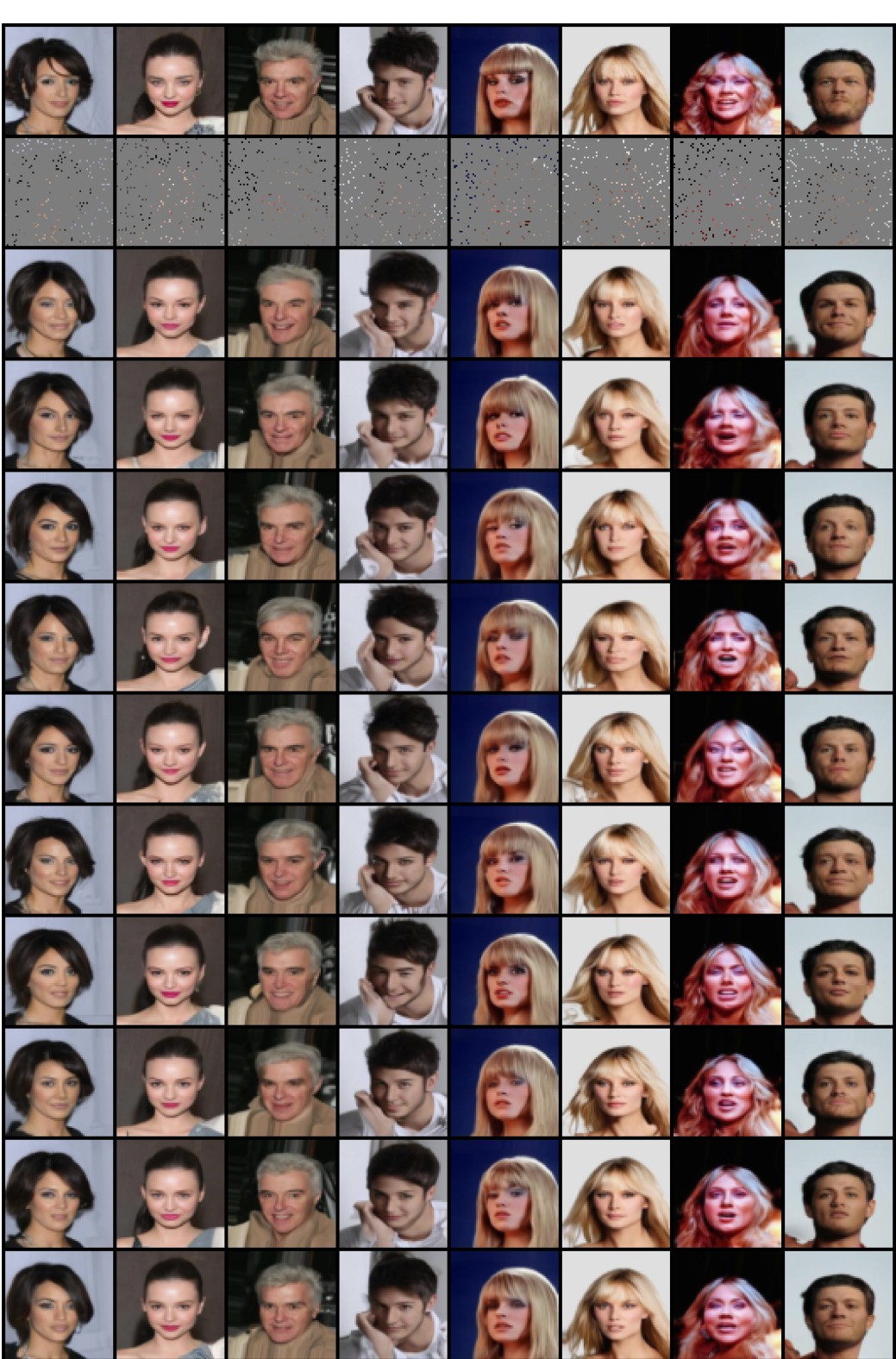

Figure 15: Additional CelebA inpainting samples.

