# OpenReview forum: "Warm Starts Accelerate Conditional Diffusion"
_ICLR.cc/2026/Conference — Submitted to ICLR 2026_

### Official Review · Reviewer_eWSh · 2025-10-28

**Soundness:** 3
**Presentation:** 3
**Contribution:** 3
**Rating:** 2
**Confidence:** 3

**Summary:**

This paper introduces a neat trick for improving the performance and efficiency of diffusion/flow models. The main idea is to first learn a conditioning-dependent Gaussian initialization (mu, std) rather than using the global distribution of (0, 1). Using samples from this distribution as a starting point, the diffusion process can be improved. A technique is introduced to transform the diffusion process such that it can still operate on the normalized space of mapping N(0,1) -> x_0. A small ablation highlights that the mean-only version of the process does not achieve the same gains, and the affect of variance normalization is crucial. Additionally, a "warmth-blending" regularizer is shown to improve performance empirically.

**Strengths:**

This paper is focused and communicates a simple, solid idea. The technique itself does not introduce significant computational overhead, as the warm-start model is set to be 10x smaller than the full diffusion model. Empirically, the effect of utilizing warm-start is a consistent improvement.

The paper is written in a clear and concise way. Figure 1 explains the intuition behind the two key ideas in the paper (using warm-start, and using the normalization transformation). Notation is clear and not overly complicated.

Experiments are conducted in a fair manner, using the same architecture and codebase.

**Weaknesses:**

- Experiments are only run on small datasets such as CIFAR. and CelebA. It is unclear how well the technique presented scales to more complex distributions such as Imagenet or text-conditioned models.
- The 'inpainting via randomly mixed pixels' task is a toy setting not used in practice, and may show outsized improvement on this specific formulation. Specifically, the randomly selected pixels give a large hint as to the mean of the resulting distribution, which may be less true for other distributions such as text or class-conditioning. This paper would be strengthened by a ablation of the performance increase when various types of conditioning is used.
- There is no related work section, which is a large issue. While the idea presented in the paper is solid, a thorough discussion of past related ideas is necessary to identify the novelty of the presented ideas.
- The introduction setting describes the problem setting, but does not describe the proposed method. It is not necessary to describe the various conditioning possibilities in the first paragraph.

**Questions:**

See above. My main ask to improve the rating of this paper is to include a thorough related work section, and secondarily to conduct an experiment of the warm-start idea to other conditioning types. Particularly, CIFAR already comes with class labels that can be used as a form of conditioning. If these points are addressed, I am willing to raise my score.

An interesting finding in this paper is that the mean-normalization aspect is less important than the variance-normalization. I wonder if there are any deeper ways to investigate this phenomenon?

---

> ### Author Response · Authors · 2025-11-20
> **Response to Reviewer eWSh**
>
> We thank the reviewer for their constructive feedback. We are encouraged that they find the idea simple, solid, and the paper clear and focused.
>
> ## Related Work Section
>
> We agree with the reviewer that a dedicated and thorough related work section would better contextualise the novelty of Warm-Start Diffusion (WSD). We already include key related works in Section 1 (Introduction), but agree that a separate Related Work section is preferable. We will expand our discussion of related works and dedicate a separate section to it in the revision.
>
>
> ## Task Scope and Conditioning Types
>
> The reviewer raises a concern that the randomly masked pixels task is a toy setting and suggests evaluating on class conditioning (e.g., CIFAR labels).
>
> **Toy Settings & Resolution:**
>
> Random pixel inpainting is a standard evaluation protocol used in vision research [1-4]. Beyond being a benchmark, it serves as a proxy for practical applications, e.g.:
> * Image super-resolution (regular low-res grid sampling)
> * Data assimilation in weather forecasting (combining sparse sensor data into complete atmospheric state)
>
> More importantly, the reviewer states:
> >Experiments are only run on small datasets such as CIFAR. and CelebA. It is unclear how well the technique presented scales to more complex distributions
>
> However, we do include a weather forecasting experiment on ERA5 data in Section 5. This is a **highly complex, real-world scientific problem operating at a much higher resolution of 240x121**, demonstrating that WSD provides strong efficiency gains on real-world datasets.
>
> We agree ImageNet experiments would strengthen the paper, but state-of-the-art ImageNet diffusion training requires ~100x more compute than CIFAR10 [5], so we prioritise ERA5 (240x121) to demonstrate scalability to high-dimensional and expensive real-world tasks.
>
> **On Class Conditioning**
>
> We appreciate this suggestion, but want to make clear that class conditioning represents the **opposite end of the conditioning spectrum** from our target applications. For class labels, the optimal Gaussian prior is analytically computable: $\hat{\mu}=$ class mean, $\hat{\sigma}=$ class std. For CIFAR-10, this yields a blurry mean image with near-uniform high variance, very close to N(0,1). This is expected: "horse" encompasses vast visual diversity, making the conditional distribution highly multimodal and poorly approximated by any single Gaussian.
>
> This is why we focus on strongly conditional tasks (inpainting, weather forecasting) where context provides tight constraints. We note that many important conditional diffusion tasks fall into this class (e.g. audio/video generation, fluid dynamics simulations, image superresolution).
>
> We acknowledge this scope limitation in Section 6 and will discuss the weakly-conditioned case in the appendix.
>
>
> ## Variance vs. Mean
>
> The reviewer asks why variance normalisation is more important than mean normalisation. We provide some intuition here, but agree that further investigation would be interesting:
>
> **Mean-only** (residual diffusion) subtracts the conditional mean, but applies uniform noise $\sigma = 1$ everywhere, ignoring which regions are well-constrained by context. For instance, in inpainting, both visible pixels and uncertain regions receive identical noise.
>
> **Full WSD** instead allows the deterministic model to appropriately pin-down highly constrained areas immediately, making it easier for the diffusion model to generate the uncertain regions. This spatial allocation of "diffusion effort" appears to be the key to low-NFE efficiency. We will clarify this in the revision.
>
>
> ## Planned Changes
>
> The reviewer stated
> > If these points are addressed, I am willing to raise my score.
>
> We believe our changes and clarifications fully address the concerns raised. Specifically, we will:
> 1. **Add a thorough Related Work section**
> 2. Clarify WSD's scope (strongly conditional tasks)
> 3. Add weakly-conditional discussion to Appendix
> 4. Expand mean-only ablation discussion
> 5. Revise the Introduction structure
>
> We hope that this encourages a reconsideration of the rating.
>
>
>
> [1] He, Kaiming, et al. "Masked autoencoders are scalable vision learners." _Proceedings of the IEEE/CVF conference on computer vision and pattern recognition_. 2022.
>
> [2] Garnelo, Marta, et al. "Conditional neural processes." _International conference on machine learning_. PMLR, 2018.
>
> [3] Gordon, Jonathan, et al. "Convolutional conditional neural processes." _arXiv preprint arXiv:1910.13556_ (2019).
>
> [4] Yu, Jiahui, et al. "Generative image inpainting with contextual attention." _Proceedings of the IEEE conference on computer vision and pattern recognition_. 2018.
>
> [5] Karras, Tero, et al. "Elucidating the design space of diffusion-based generative models." _Advances in neural information processing systems_ 35 (2022): 26565-26577.

---

### Official Review · Reviewer_UpBA · 2025-10-29

**Soundness:** 2
**Presentation:** 1
**Contribution:** 2
**Rating:** 2
**Confidence:** 4

**Summary:**

This paper proposed the warm-up method to accelerate the sampling speed of the diffusion model. The key motivation of this paper is that we can train a model to generate a better start point for diffusion models, thereby reducing the unnecessary timesteps. The experimental results demonstrate that the proposed method can accelerate the diffusion models.

**Strengths:**

1. The overall method is reasonable since a good starting point, e.g., a gold seed, can bootstrap the diffusion model`s generation speed.

**Weaknesses:**

1. The overall writing of this paper should be further improved. For example, 1) typos. See 118 lines. 2) Section 2 is too confusing. Section 2.3 should be removed from Section 2 and moved to a new section to illustrate how to train WSD in multi-task settings.

2. The overall method is unconvinable. To begin with, this paper claims that WSD is model-agnostic. However, WSD needs to first generate a new dataset from its output. How can this situation be model-agnostic? Meanwhile, each diffusion model needs to fine-tune on this new dataset, thereby leading to a higher computational cost than direct model distillation. The overall pipeline contains: 1) First training WSD, 2) then fine-tuning diffusion models, which is unconvinable.

3. The experimental results lack the necessary baselines. This paper has no baseline to compare against. For example, the warm start is similar to generating good seeds. Therefore, these works can serve as a baseline [1]. Then, the training cost of this paper seems to reach the distillation. How to illustrate that WSD is better than distillation?

[1] Golden Noise for Diffusion Models: A Learning Framework. Zhou et al. ICCV 2025.

**Questions:**

No question, please see Sec. Weaknesses part.

To sum up, this paper proposed WSD as a good starting point for diffusion models. But due to the lack of necessary baselines and the overall method's unconvincing nature, this paper is obviously below the bar for acceptance. I rate it as reject.

---

> ### Author Response · Authors · 2025-11-19
> **Response to Reviewer UpBA**
>
> We thank the reviewer for their comments. We believe the reviewer’s concerns stem from several misunderstandings of the method and experimental setup.
> ## On Baselines
>
> The reviewer states that the paper "has no baseline to compare against". Contrary to the reviewer’s claim, the paper evaluates against a strong, state-of-the-art sample-efficient baseline (Sec. 3), combining flow-matching with DPM-Solver++. Our method combines this strong baseline with the warm-start approach, making the two directly comparable.
>
> The "Golden Noise" method the reviewer suggests as a baseline is **entirely orthogonal** to warm-start diffusion, as it optimises the **seed** used for sampling noise, whereas WSD optimises the prior **distribution** from which noise is sampled.
>
> We also point out that ICCV 2025 was less than 4 months ago, making the "Golden Noise" paper concurrent work under the ICLR policy, so it is not expected to be included as a baseline.
>
> We address distillation as an alternative efficient method in Section 1. WSD differs significantly in that it is a small modification of standard diffusion (and therefore compatible with most diffusion methods), whereas distillation is a significantly more complex, unstable, and expensive modelling paradigm.
>
> ## On Model-Agnosticism
>
> >1. The overall method is unconvinable. To begin with, this paper claims that WSD is model-agnostic. However, WSD needs to first generate a new dataset from its output. How can this situation be model-agnostic? Meanwhile, each diffusion model needs to fine-tune on this new dataset, thereby leading to a higher computational cost than direct model distillation. The overall pipeline contains: 1) First training WSD, 2) then fine-tuning diffusion models, which is unconvinable.
>
> The reviewer states that WSD requires generating a dataset and fine-tuning diffusion models, but **these steps do not appear anywhere in our method or algorithms**. The method works by applying conditional normalisation, which is computed on-the-fly during training and inference. The method is model-agnostic because it is compatible with any standard diffusion-style generative model (DDPM, DDIM, Flow Matching, EDM, etc.). We refer the reviewer to Algorithm 1 (training) and Algorithm 2 (inference), which clearly outline the method and its model agnosticism.
>
> The computational overhead of training the warm-start model amounts to less than 10% of total training compute, making it significantly cheaper than any distillation-based approaches.
>
> ## On Writing and Multitask Training
>
> The reviewer notes that the overall writing of the paper should be improved. We thank the reviewer for the typo on line 118, but this is the only mistake that is mentioned, not providing any other actionable examples that would justify a "poor" rating for presentation.
>
> The reviewer also asks us to restructure the paper
> > Section 2 is too confusing. Section 2.3 should be removed from Section 2 and moved to a new section to illustrate how to train WSD in multi-task settings.
>
> The warmth-blending mechanism discussed in Section 2.3 is about an enhanced training method that effectively modifies the WSD training to be over different augmentations of the dataset (parameterised by the warmth $w$). This can be viewed as a form of multi-task training, but is unrelated to multi-task **settings**. We will clarify this section in the revision, but believe that it belongs in Section 2 (Method), as it is an important part of the method.
>
> We are grateful for any other sources of confusion that the reviewer has found, so that we can improve the clarity of the paper.
> ## Planned Changes
>
> 1. Fix the typo outlined on line 118
> 2. Clarify the language in Section 2.3, avoiding further confusion.
>
> We hope the above clarifications help address the reviewer’s concerns.

---

> > ### Comment · Reviewer_UpBA · 2025-11-28
> >
> > Thanks to the author`s rebuttal, I have carefully checked all the contents, including the revised manuscript. This improved version addresses many concerns regarding the motivation for the proposed WSD. Firstly, the revised manuscript clarifies the motivation and provides more details about the WSD. This addresses my concerns about the inconveniences for WSD.
> >
> > Then, there are some concerns: 1) The insufficient experiments. No baseline is reported in the Figs. 5-6. In my view, WSD is similar to the mid-training methods that leverage a light-aware model with a cheaper training process to improve the diffusion model. In that case, some similar mid-training approaches, such as gold seeding, should be considered to demonstrate that WSD is better, even though WSD is totally orthogonal to these methods. At least, the author should report how much FID is gained after building WSD upon them.
> >
> > Meanwhile, I have checked the official rule of ICLR:
> >
> > Q: Are authors expected to cite and compare with very recent work?
> >
> > A: We consider papers contemporaneous if they are published within the last two months. That means, since our full paper deadline is September 24, if a paper was published (i.e., at a peer-reviewed venue) on or after July 24, 2025, authors are not required to compare their own work to that paper.
> >
> > Note that the notification date for ICCV 25 is Jun 25 '25, so the author should compare within that scope.
> >
> > 2) The author may overclaim the score of the task that WSD can work on. Figure 1 shows almost six types. But the experiments only report performance on three types: image inpainting/denoising and weather forecasting. This cannot support the claim of the paper.
> >
> > To sum up, the revised manuscript addresses my concerns about the inconveniences for WSD. But there are concerns about the experiments. Therefore, I temporarily increased my score to marginally below the acceptance threshold.

---

> > > ### Author Response · Authors · 2025-11-28
> > > **Response to Reviewer UpBA**
> > >
> > > We thank the reviewer for their comments. We are pleased to hear that the revised manuscript has resolved some of the misunderstandings, and that the reviewer has updated the score accordingly.
> > >
> > > We now address the reviewer's remaining concerns.
> > >
> > > ### 1) Baselines
> > >
> > > The reviewer suggests that our experiments are insufficient, specifically asking us to include a baseline in Figures 5 and 6. We ask the reviewer to review these figures, because they **do include a baseline** (Flow Matching + DPM Solver). In both figures, we show that WSD requires a smaller NFE to achieve the same performance / a better performance than the baseline
> > >
> > > We show that WSD is compatible and synergistic with efficient solvers like DPM Solver, as it has become the de-facto standard for efficient sampling. A quantitative **comparison** with other methods only makes sense if these methods were incompatible with WSD, i.e. also attempting to change the noise distribution based on the context information. Otherwise, we could always design a task in which WSD is more effective (very strong constraints), or a task in which WSD is less effective (weaker constraints), which would be misleading.
> > >
> > > **Golden Noise**
> > >
> > > The reviewer again suggests we compare WSD to the golden noise method [1]. We thank the reviewer for clarifying the ICLR policy on contemporaneous work, which has changed since 2025. Our previous claim was based on this outdated policy. We acknowledge that [1] falls within the comparison scope.
> > >
> > > However, we still believe that comparing WSD to [1] is still not a useful comparison for the following reasons:
> > > - Domain mismatch: [1] only considers text-to-image generation, a domain that is specifically out of scope for WSD. Adapting the method from [1] to a completely different domain (like image inpainting or weather forecasting) only to test how these methods interact is complicated and out of scope for WSD.
> > > - Goal mismatch: The golden noise method **slightly increases inference time compute** to improve image **quality**, whereas WSD **substantially reduces inference time compute** to improve **sampling efficiency**. WSD also slightly improves high-NFE performance, but this is not its main goal.
> > > - Orthogonality: As the reviewer noted,  the two methods are orthogonal: [1] adjusts the seed to achieve higher generation quality, whereas WSD changes the prior **distribution** to improve sampling speed. They are not competing ideas, and demonstrating their synergy would require a large engineering effort due to the domain mismatch.
> > >
> > > ### 2) Scope of Claims
> > > The reviewer suggests that our paper overclaims applicable domains. We clarify that Figure 1 illustrates the **class of problems** (strongly conditional generation) where WSD is most likely to work well, rather than a checklist of experiments. However, our experiments were carefully chosen to be similar to these tasks:
> > > - Inpainting is very similar to denoising/super-resolution (restoring missing information based on fixed available pixels).
> > > - Weather forecasting is similar to fluid dynamics predictions/video generation (autoregressive spatiotemporal modelling, strongly constrained by the previous frame).
> > >
> > > Based on these similarities and the generality of our method, we believe that WSD is applicable to these closely related domains. However, we never claim to prove that this is the case empirically.
> > >
> > > We hope this addresses the reviewer's remaining concerns and justifies the applicability of our method.
> > >
> > > [1] Golden Noise for Diffusion Models: A Learning Framework. Zhou et al. ICCV 2025.

---

### Official Review · Reviewer_AiWQ · 2025-10-29

**Soundness:** 3
**Presentation:** 2
**Contribution:** 2
**Rating:** 4
**Confidence:** 4

**Summary:**

This paper introduces "Warm-Start Diffusion (WSD)," a method designed to accelerate the sampling process in conditional diffusion models. Traditional diffusion models typically start sample generation from an uninformed Gaussian noise distribution, requiring a large number of function evaluations (NFEs). WSD addresses this by incorporating a simple, deterministic "warm-start" model that predicts the initial moments (mean and standard deviation) of the conditional data distribution, thereby providing an "informed" prior closer to the true data distribution. This significantly reduces the distance the generative process must traverse, leading to a substantial decrease in the required number of diffusion steps. WSD is compatible with any standard diffusion or flow matching algorithm and can be combined with other fast sampling techniques. The paper evaluates WSD on image inpainting and weather forecasting tasks, demonstrating that it generates high-quality samples with only 4-6 function evaluations and saturates performance with 10-12, significantly outperforming traditional methods. Additionally, the paper proposes a "conditional normalisation trick" and a "warmth blending" mechanism to further enhance the method's effectiveness and compatibility.

**Strengths:**

1. Sampling efficiency improvement. WSD drastically reduces the number of function evaluations (NFEs) required to generate high-quality samples. It achieves high-quality image generation with as few as 4-6 NFEs and saturates performance around 10-12 NFEs, which is crucial for practical applications requiring fast sample generation (e.g., in autoregressive generation tasks).
2. Modular design. WSD acts as a "plug-and-play" method that can be combined with any standard diffusion or flow matching algorithm without extensive re-derivation or major modifications to existing models. This makes the method flexible and broadly applicable.
3. The normalisation trick enables WSD to seamlessly integrate with existing diffusion algorithms that assume noise sampling from a standard Gaussian distribution, avoiding complex re-implementation efforts.

**Weaknesses:**

1. Gaussian prior assumption of the warm-start model. WSD's warm-start model assumes an uncorrelated Gaussian posterior for the conditional data distribution. While effective for tasks with strong conditioning information (like image inpainting or weather forecasting), a single Gaussian may be insufficient to capture complex distributions in highly multimodal or weakly conditioned settings (e.g., text-to-image generation), limiting its utility in such tasks.
2. Task/Dataset dependency of the model. Currently, a separate warm-start model needs to be trained for each experiment and dataset. Although the paper mentions the possibility of training a single general-purpose warm-start model, this remains a limitation that adds complexity and cost to practical applications.
3. Less pronounced gains at high NFEs. While WSD excels at low NFEs, its performance gains compared to standard flow matching are less significant in the high-NFE saturation regime. This implies that the benefits of WSD are limited when computational resources are abundant.
4. Introduction of additional model complexity. Although introducing an additional warm-start model improves efficiency, it also increases the overall model complexity, including an extra training phase and additional parameters.

**Questions:**

1. The optimization loss of the first stage.
2. The effectiveness on other tasks (e.g., image editing) and higher resolutions.

---

> ### Author Response · Authors · 2025-11-19
> **Response to Reviewer AiWQ**
>
> We thank the reviewer for their thoughtful evaluation. We are glad to hear that they appreciate the method's significant improvement in sampling efficiency, its modular design, and the effectiveness of the conditional normalisation trick. The reviewer rated our presentation a 2 (Fair), and we welcome any additional guidance on how it might be improved.
>
> We address the specific concerns and questions below.
>
> ### On the Gaussian Prior Assumption
>
> The reviewer notes that the warm-start model assumes a single Gaussian posterior, which may be insufficient for multimodal distributions. We agree that this is a limitation, but highlight that standard diffusion **also relies on a diagonal Gaussian prior**, $\mathcal{N}(0, I)$. This is a **strictly worse approximation** of the conditional distribution than the informed prior provided by our warm-start model, $\mathcal{N}(\hat{\mu}_C, \hat{\sigma}_C)$. Effectively, standard diffusion is a special case of WSD where the warm-start model is forced to predict $\hat{\mu}_C = 0, \hat{\sigma}_C = I$.
>
> We also note that a large class of important diffusion tasks (video/audio-generation, weather forecasting, super-resolution, colorisation etc.) have strong conditioning information available, and represent classes of problems in which computational efficiency is extremely useful.
>
> ### Task Dependency and Added Complexity
>
> The reviewer notes that training a separate warm-start model adds complexity and requires task-specific training.
>
> While our method adds a ~10% overhead in parameter count and training compute, we believe this to be a favourable trade-off for the observed 2x speedup in sampling. Allocating a similar increase in parameters to a standard diffusion model would typically yield only modest improvements, especially at low NFE.
>
> Regarding deployment complexity, we suggest viewing the warm-start model and the generative model as a single unit with detached gradients between the two stages. This combined model can be trained, stored, and deployed as one entity.
>
> The fact that WSD is task-dependent and therefore requires training on the target data is consistent with the baselines we compare against, and standard practice for generative models.
>
> ### High NFE Saturation
>
> The reviewer correctly notes that gains are less pronounced at high NFE. We acknowledge this, but note that the primary motivation of WSD is to unlock performance in the **efficient sampling regime** (low NFE), which is critical for computationally intensive applications like autoregressive weather forecasting or video generation.
>
> ### Questions
>
> **1. The optimisation loss of the first stage.**
>
> The warm-start model is trained using the Gaussian Negative Log-Likelihood (NLL) loss, as detailed in Equation 5 of the paper. This acts as a standard probabilistic regression objective, encouraging the model to predict the mean and marginal variance of the conditional data distribution.
>
> **2. The effectiveness on other tasks (e.g., image editing) and higher resolutions.**
>
> - **Image editing:** We anticipate that WSD would be effective for image editing tasks (e.g., inpainting, super-resolution, colorisation). These are typically strongly conditional tasks where the output is constrained by the input image, meaning the conditional distribution is well-approximated by the warm-start model's predictions.
> - **Higher resolutions:** Our weather forecasting experiment (240x121) already operates at a significantly higher dimensionality than CIFAR-10/CelebA, and WSD proved highly effective there. We currently do not have the computational resources to train on a large high-resolution dataset like Imagenet, and hope the existing results on the ERA5 data sufficiently demonstrate WSD's performance in high-resolution tasks.
>
> ### Planned Changes
>
> Based on this feedback, we will:
> 1. Clarify in the manuscript that WSD degrades gracefully in multimodal settings, as it is strictly superior to the $\mathcal{N}(0, I)$ prior.
> 2. Explicitly frame the warm-start and generative models as a single trainable/deployable unit to address complexity concerns.
>
> Given the reviewer’s positive assessment that the method is important for practical applications and helps substantially reduce NFEs, we would be grateful if the reviewer might consider revisiting their score in light of these clarifications.

---

### Official Review · Reviewer_AevA · 2025-10-31

**Soundness:** 3
**Presentation:** 3
**Contribution:** 2
**Rating:** 4
**Confidence:** 4

**Summary:**

This paper introduces Warm-Start Diffusion (WSD), a method to accelerate sampling in conditional diffusion and flow-matching models. Standard diffusion starts from an uninformed noise prior, $\mathcal{N}(0, I)$. WSD instead uses a small, separate, deterministic model ($h_{\phi}$) to predict an *informed* prior, ${N}(\mu_{C}, {\sigma}_{C}^{2} ) $,

based on the conditional context $C$. By starting from this "warm" state, which is closer to the target data distribution, the diffusion process has a shorter distance to traverse, thereby reducing the required Number of Function Evaluations (NFE). The method uses a "conditional normalisation trick" to train a standard generative model $p_{\theta}^{\prime}$ in a normalized space, making WSD compatible with existing diffusion frameworks. Experiments on image inpainting and ERA5 weather forecasting demonstrate that WSD can generate high-fidelity samples in 4-12 NFE, substantially outperforming a standard flow-matching baseline in the low-NFE regime.

**Strengths:**

* **Modular and Simple:** The proposed method is straightforward to implement. It consists of two distinct components: a warm-start regression model ($h_{\phi}$) and a generative model ($p_{\theta}^{\prime}$). This modularity allows any suitable Gaussian regression model to be used for $h_{\phi}$ without retraining.
* **Compatible and Synergistic:** WSD is orthogonal to other sampling acceleration techniques, such as efficient ODE solvers. The paper demonstrates this by combining WSD with a flow-matching model and a DPM-Solver.

* **Strong Ablation Studies:** The ablations (Sec 4.2) effectively validate the design choices. The "Mean-only" ablation confirms that predicting the conditional standard deviation  is the critical component for low-NFE gains, not just predicting the mean (residual diffusion). The "Feature only" ablation confirms that the performance benefit stems from the warm-start normalization itself.

* **Relevant Application:** The method is tested on tasks like autoregressive weather forecasting, where sampling efficiency is a primary practical bottleneck, making the work relevant.

**Weaknesses:**

* **Limited Applicability:** The method's core assumption is that the conditional posterior $p(X_0|C)$ can be reasonably approximated by a *single, unimodal Gaussian with a diagonal covariance matrix* ($\mathcal{N}(\hat{\mu}_{C}, \text{diag}(\hat{\sigma}_{C}^{2}))$). This is a severe limitation. The method only works for **strongly conditional** tasks (e.g., inpainting, 6-hour weather steps) where the output is already highly constrained. The claim of being "widely applicable" is an overstatement. The method will fail on weakly conditional or multimodal tasks (like text-to-image), where this prior is entirely insufficient.

* **Ad-Hoc Fix for High-NFE:** The paper reports that the base WSD method *underperforms* standard flow matching in the high-NFE regime. The proposed "warmth blending" (Sec 2.3) is an ad-hoc fix for this. This underperformance implies the learned warm-start prior is fundamentally flawed—likely too confident and simplistic (due to the diagonal $\hat{\sigma}_{C}$)—and acts as an incorrect constraint. The multi-task blending just papers over this structural problem.

* **Added Overhead:** WSD introduces the overhead of training, storing, and running a *second* model, $h_{\phi}$. Although the authors state $h_{\phi}$ is small and do not count its single forward pass in the NFE, it is still an added computational step. Furthermore, this warm-start model is task-specific and must be trained from scratch for each new dataset.

* **Simplistic Prior:** The reliance on a diagonal covariance matrix ignores all correlations in the data. This is a crude approximation, likely responsible for the poor high-NFE performance and limiting the method's ultimate sample quality.

**Questions:**

1.  The "warmth blending" (Sec 2.3) was introduced because WSD performs worse than the baseline at high NFE. Does this not point to a fundamental failure of the warm-start model's assumptions? If the $\mathcal{N}(\hat{\mu}_{C}, \text{diag}(\hat{\sigma}_{C}^{2}))$ prior was a *good* approximation, performance should saturate *above* the baseline, not below it. Why is this behavior not interpreted as evidence that the diagonal Gaussian prior is simply incorrect?

2.  In your "Mean-only" ablation, you show that predicting $\sigma_{C}$ is key to low-NFE performance. This suggests the generative model $p_{\theta}^{'}$ learns to ignore regions with low predicted variance. In Figure 2a, the $\sigma_{C}$ image looks like a simple blur. Does $\sigma_{C}$ learn anything more sophisticated than "high variance for missing pixels, low variance for visible pixels"? For example, does it capture semantic uncertainty (e.g., higher uncertainty on a mouth vs. a cheek)?

3.  The method is only tested on strongly conditioned tasks. How does WSD's performance degrade as this conditioning $C$ is weakened? For the inpainting task, what happens to the FID vs. NFE curve if you provide 50% or 80% of the pixels, rather than just 5-10%? At what point does the unimodal Gaussian prior become useless and the method's advantage vanish?

---

> ### Author Response · Authors · 2025-11-19
> **Response to Reviewer AevA**
>
> We thank the reviewer for their detailed and insightful assessment. We particularly appreciate the recognition of the method’s modularity and the strength of the ablation studies.
>
> ### On the "Simplistic Prior" and Warmth Blending (Weaknesses & Q1)
>
> The reviewer suggests that the necessity of warmth blending indicates the warm-start prior is "fundamentally flawed." We respectfully offer a different perspective. It is important to note that while the diagonal Gaussian prior $\mathcal{N}(\hat{\mu}_C, \hat{\sigma}_C)$ is a crude approximation, standard diffusion **also relies on a diagonal Gaussian prior**, $\mathcal{N}(0, I)$. This is a **strictly worse approximation** of the conditional distribution than the informed prior provided by our warm-start model. Standard diffusion can effectively be viewed as a special case of our method where the warm-start model is forced to predict $\hat{\mu}_C = 0, \hat{\sigma}_C = I$. If the "incorrect" $\mathcal{N}(0, I)$ prior does not inhibit standard diffusion, we argue that our informed prior should not either.
>
> Regarding the high-NFE performance, we hypothesise that the base WSD method underperforms because diffusion models generally struggle with heavy-tailed distributions [1]. Normalising the data removes Gaussian components, leaving highly non-Gaussian data. We believe warmth blending allows the model to learn across a range of tasks, from the Gaussian ($w=0$) to the normalised ($w=1$) setting, and forces the model to explicitly learn this transformation.  We acknowledge that further work is required to fully understand this mechanism, but empirical results demonstrate that warmth blending is effective.
>
> ### Applicability and Weak Conditioning (Weaknesses & Q3)
>
> We agree that our method yields the most significant gains in strongly conditional tasks. We used the term "widely applicable" to refer to domains like weather forecasting, fluid dynamics, and video generation. In these settings, computational costs are high because the cost of a single sample is multiplied by autoregressive rollouts and ensemble members, making the efficiency of WSD extremely valuable.
>
> Regarding Q3, we clarify that WSD does not "fail" on weakly conditional tasks but **gracefully degrades**. Since the warm-start model minimises Gaussian NLL over the training distribution, it learns parameters that are at least as good as the fixed $\mathcal{N}(0, I)$ prior. In the limit of no conditioning, it would learn the dataset's pixel-level statistics, converging to (not below) standard diffusion performance.
>
> For stronger conditioning (e.g., 50-80% of pixels), the conditional distribution can be well-approximated by a narrow unimodal Gaussian, so we expect the deterministic model to be highly accurate, likely yielding even bigger efficiency gains over standard diffusion.
>
> A comprehensive study of performance across the full spectrum of conditioning strengths would be valuable future work, but each setting would require the training of a new model, which may be impractical during the review timeline because of limited compute.
>
> ### Added Overhead
>
> While our method adds a ~10% overhead in parameters and compute, we believe this is a favourable trade-off for the dramatic speedup. If one were to instead allocate 10% more parameters to a standard diffusion model, the performance gain would be marginal compared to the 2x efficiency improvement provided by WSD.
>
> Regarding the additional separate models, it may be better to frame the warm-start model + generative model as a single model with detached gradients. This combined model can be trained, stored, and run as one unit. This model needs to be trained separately for each task, but so does a standard generative model.
>
> ### Visualising Sigma (Response to Q2)
>
> Upon inspection of $\hat{\sigma}_C$ in Figure 2a, we do not find that it appears as a "simple blur." The model clearly learns spatially varying uncertainty: higher values at facial features (eyes, nose, mouth), hair-face boundaries, and sparsely-conditioned regions (right neck). This aligns with our mean-only ablation (Sec 4.2), which demonstrated that predicting $\hat{\sigma}_C$ is critical for low-NFE performance. To make this more apparent in the revision, we will use a colormap with better contrast.
>
> ### Planned Changes
>
> 1. Use more precise language around applicability (strongly conditional tasks)
> 2. Clarify the expected behaviour across conditioning strengths (Question 3).
> 3. Add discussion on the "heavy-tailed residual" hypothesis surrounding warmth-blending, while noting that further investigation is needed.
> 4. Update $\hat{\sigma}_C$ in Figure 2a with a higher-contrast colourmap.
> 5.  Clarify that the warm-start model and generative model can be
>    trained, stored, and deployed together as a single unit.
>
> [1] Pandey, K., Pathak, J., Xu, Y., Mandt, S., Pritchard, M., Vahdat, A., & Mardani, M. (2024). Heavy-tailed diffusion models. _arXiv preprint arXiv:2410.14171_.

---

### Meta-Review · Area_Chair_Q2KC · 2026-01-02

**Summary:**

This paper proposes Warm-Start Diffusion (WSD), a method for strong-condition diffusion models that replaces the standard Gaussian initialization with learned, favorable initial states. The core idea is well-motivated, and the authors provide extensive experiments demonstrating the effectiveness and generalization of WSD under various strong-conditioning settings—particularly in image editing tasks.

However, all reviewers raised consistent concerns regarding the practical applicability and broader validity of the WSD. While the method shows notable improvements in specific strong-condition scenarios, its generalizability has not been sufficiently validated. In particular, the evaluation lacks experiments on large-scale benchmarks (e.g., ImageNet) and common unconditional or weakly conditioned generative settings, which are essential to assess the method’s robustness and wider utility.

Given these limitations and the reviewers’ overall assessments, the paper cannot be accepted in its current form. We encourage the authors to address these concerns—especially by expanding the experimental validation—and to clarify the scope and assumptions underlying the proposed approach.

**Reviewer Concerns:**

All reviewers raised consistent concerns regarding the practical applicability and broader validity of the WSD. While the method shows notable improvements in specific strong-condition scenarios, its generalizability has not been sufficiently validated. In particular, the evaluation lacks experiments on large-scale benchmarks (e.g., ImageNet) and common unconditional or weakly conditioned generative settings, which are essential to assess the method’s robustness and wider utility.

**Reviewer Scores:**

Had the reviewer been able to fully participate in the discussion, I believe their score would likely have remained largely unchanged. I appreciate the feedback provided and will carefully address these points in a revised version of the manuscript.

---

### Decision · Program_Chairs · 2026-01-26

Reject